# A Scoping Review of the Application of Metabolomics in Nutrition Research: The Literature Survey 2000–2019

**DOI:** 10.3390/nu13113760

**Published:** 2021-10-24

**Authors:** Eriko Shibutami, Toru Takebayashi

**Affiliations:** 1Graduate School of Health Management, Keio University, Kanagawa 252-0883, Japan; eshibutami@keio.jp; 2Department of Preventive Medicine and Public Health, Keio University School of Medicine, Tokyo 160-8582, Japan

**Keywords:** metabolomics, nutrimetabolomics, biomarker, nutrition, food, diet, dietary pattern

## Abstract

Nutrimetabolomics is an emerging field in nutrition research, and it is expected to play a significant role in deciphering the interaction between diet and health. Through the development of omics technology over the last two decades, the definition of food and nutrition has changed from sources of energy and major/micro-nutrients to an essential exposure factor that determines health risks. Furthermore, this new approach has enabled nutrition research to identify dietary biomarkers and to deepen the understanding of metabolic dynamics and the impacts on health risks. However, so far, candidate markers identified by metabolomics have not been clinically applied and more efforts should be made to validate those. To help nutrition researchers better understand the potential of its application, this scoping review outlined the historical transition, recent focuses, and future prospects of the new realm, based on trends in the number of human research articles from the early stage of 2000 to the present of 2019 by searching the Medical Literature Analysis and Retrieval System Online (MEDLINE). Among them, objective dietary assessment, metabolic profiling, and health risk prediction were positioned as three of the principal applications. The continued growth will enable nutrimetabolomics research to contribute to personalized nutrition in the future.

## 1. Introduction

Nutrimetabolomics is an emerging field in nutrition research, and it is expected to play a significant role in deciphering the interaction between diet and health as it continues to develop [1,2,3,4,5]. Nutrition research has historically focused on individual nutrients and specific foods in terms of their risk to human health. However, to understand the full role of nutrients in the body, it is necessary to clarify the relationship between the intricately intertwined dietary and biological factors that affect the uptake and utilization of nutrients. Through the growth of “omics” technology since the beginning of this century, the definition of food and nutrition has changed from sources of energy and macro/micro-nutrients to an essential exposure factor to determine health risks. Furthermore, this new approach has enabled nutrition research to identify biomarker candidates of several dietary factors as well as to deepen the understanding of metabolic dynamics and impacts on health risks.

Omics science aims to analyze the interaction and function of large amounts of biological information and to understand how they contribute to human health [6]. Metabolomics is one of the principal elements in systems biology. The measurement subjects are low-molecular compounds (molecular weight of up to approximately 1500) contained in biological fluids, such as blood and urine, mainly targeting sugars, amino acids, organic acids, peptides, fatty acids, and their analogues. Initially, the concept of “metabonomics” was defined by Nicholson in 1999 [7] as the first method to apply system biology to study the metabolism; that is, “the quantitative measurement of the dynamic multiparametric metabolic response of living systems to pathophysiological stimuli or genetic modification.” Within this concept, “metabolomics” mainly focuses on the metabolic profiling of cells and organs, defined as “the comprehensive analysis of the whole metabolome (all metabolites synthesized by an organism)” by Fiehn in 2002 [8]. The terminology “nutrimetabolomics” was introduced by Zhang et al. [9] in their review article as one of the omics techniques to be used in nutritional research.

Against this backdrop, nutrimetabolomics is expected to contribute to the application of precision nutrition to disease prevention and management in the future, as the field has the potential to predict individual metabolic dynamics with objective indicators [10,11]. However, so far, candidate markers identified by nutrimetabolomics have not been clinically applied. One of the reasons for the stagnation may be that the rapid development of the field has made researchers short-sighted and ad hoc. Therefore, a big picture perspective is required to realize the future of precision nutrition.

Thus, the purpose of this scoping review was to provide a bird’s-eye view of the trajectory and extent of dietary biomarker development in order to help nutrition researchers better understand the potential of its application, including the historical development, recent focuses, future prospects, and issues that remain to be overcome. To achieve this goal, we conducted an extensive literature review of human studies in the field of nutrimetabolomics published over the last two decades to show the evolutionary trends of this field based on the number of articles and notable studies.

## 2. Materials and Methods

To provide the outline of the growth of nutrimetabolomics, we carried out an extensive literature search exploring human studies published in the past twenty years.

### 2.1. Search Strategy

The literature search strategy shown in Table 1 targeted human studies published in English that were available between 1 January 2000 and 31 December 2019, contained in the electronic database; the Medical Literature Analysis and Retrieval System Online (MEDLINE) via PubMed. The study records were identified using the following two groups of search terms: (1) metabolome approaches: “metabolomics” OR “metabonomics”; (2) dietary factors: “nutrition” OR “food” OR “diet” OR “meal” OR “intake” OR “consumption”, and an AND search was performed on the two groups. These keywords were converted to PubMed-defined transition terms including MeSH in the database search formula (see Appendix A for details of the database search formula and transition terms). The literature search was conducted on 23 February 2021.

### 2.2. Selection Criteria

Studies for which the full text was not available, reviews, conference reports, and study protocols were excluded. Non-human researches (plant and food, animal, and cell researches), measurement methods and algorithms, non-metabolomics field researches (researches of single-biomarkers or non-exhaustive metabolites, other omics researches, and researches that were within the scope of general clinical testing), and metabolomics researches whose main purpose was other than dietary factors (disease, diagnosis, exercise, toxicity, drugs, and environment) were also excluded.

### 2.3. Classification and Data Collection

The selected studies were categorized into study designs, biofluids, application fields, and dietary factors. Furthermore, the categorized dietary factors were subcategorized separately, and the targeted health risks were classified by disease group (details of the classification are shown in Appendix A). Additional information was also collected, including the number and gender of subjects, measurement methods, and integrated epidemiological research projects. The proportions of the total number of articles from 2000 to 2019 and the annual transition from 2010 to 2019 were charted by category, and the transition was analyzed for the number and percentage. For the definition of biomarkers, the classifications of Gao [12] were referred to. The article extraction, classification, and charting were conducted by E.S., and T.T. supervised the entire search strategy and analysis.

## 3. Results and Discussion

In this scoping review, we aggregated the number of human metabolomics studies focused on nutrition and diet to clarify the association between dietary factors and health risks. Initially, the search engine identified 3112 records from the database. From the records, we extracted 502 studies that met the criteria by screening the titles and abstracts of records, halting those that could not be judged from the abstract. The full text was evaluated to further verify that the articles met the criteria, and finally 452 studies were included in this review (the flow of search and selection is shown in Figure 1).

Since the beginning of the 2000s, metabolomics research focused on nutritional and dietary factors had gradually begun to be applied to human studies. Thereafter, the number of reports exponentially increased from 2010, and in 2019, 114 research articles were published, approximately 70% more than the number published in the previous year (Figure 2). The high expectations for nutrimetabolomics in nutritional research have been demonstrated by this rapid increase in the number of reports in the last 10 years, taking advantage of the introduction of high-sensitivity detection methods such as mass spectrometry (MS) in addition to the initial nuclear magnetic resonance (NMR) [3,4].

Looking at the development of this field, the topics of interest have changed throughout the early (I 2000–2009), middle (II 2010–2014), and recent (III 2015–2019) periods. Particularly in the recent period, there have been signs of growth in studies applying dietary pattern and disease risk prediction. Summaries of the notable trends are shown in Table A1 and Table A2 (details of aggregated results are shown in Appendix A).

### 3.1. Pioneering Studies (2000–2009)

Before the concept of metabonomics was defined by Nicholson in 1999, there was a pioneering observational study of dietary factors and metabolites by Zuppi et al. (1998) [13], which reported a comparison between different populations in Rome and at arctic latitudes in Svalbard. Following this landmark study, during the early stages of nutrimetabolomics research between 2000 and 2009, only a few studies were published each year. Studies during the early period mainly analyzed urine samples with NMR by non-randomized clinical trials (NRCTs) or crossover studies of randomized controlled trials (RCTs) for small-scale populations; then, studies with blood samples by MS-based methods in larger studies increased in the later years (Table 2). Some focused on basic fluctuation in biofluids that underpinned subsequent studies [14,15,16], and others explored on beverages and foods, which are rich in phytochemicals including various types of tea [17,18], coffee [19,20], and cocoa [21], as well as phytochemicals themselves [22,23].

The pioneering researchers initially had to examine the influences of lifestyle and individual variation on metabolite changes using biofluids to determine the feasibility of the analysis and the data-comparability for studies conducted in different regions. Lenz [14] reported that urinary samples demonstrated relatively large inter-individual variability, while plasma samples showed small inter/intra-individual variability. Their further comparative study of biofluids [15] clarified the difference in the reactivity of metabolic changes in food intake between participants in the United Kingdom and Sweden, revealing that metabolic profiling was susceptible to distinct cultural and extreme diets. Walsh [16] also examined the responsiveness of the metabolic profile to acute dietary effects using urine, plasma, and saliva, highlighting the importance of adjusting factors such as recent dietary intake or time of sample collection, in protocol planning.

### 3.2. Study Design

Intervention studies accounted for two-thirds of the total, while observational studies accounted for the rest (Figure 3a). 

The number of subjects in intervention had a median (IQR) of 30 (15‒59), with the largest study having 983 subjects, which was related to the Mediterranean diet (MED) and cardiovascular disease (CVD) in the Prevención con Dieta Mediterránea (PREDMED) study [32], while the number of subjects in the observational studies had a median (IQR) of 229 (70‒654) with the largest study having 5620 subjects, which was a cross-sectional research related to coffee consumption and the thyroid function in the Inter 99 study [33].

The most common study design was RCT parallel-group comparison studies, accounting for 28% of the total. In addition, since the washout period is generally short in dietary intervention tests, RCT crossover studies (25%) that could be applied to a small number of subjects were also pragmatic choices. As sample sizes in nutritional intervention remain relatively small and are usually implemented in a controlled setting, the identified dietary biomarkers cannot directly be extrapolated to free-living individuals. Thus, to detect associations between metabolites and disease, epidemiological studies with large but attainable sample sizes may be required.

Observational studies are useful designs for investigating metabolite changes under habitual diet. Cross-sectional studies, including cohort baseline studies, accounted for 18% of the total, whereas cohort follow-up studies were a few (5%). Guertin [5] proposed the required sample size of 80% power to detect an association in a 1:1 case-control study; for a large effect, samples of around 200–400 individuals would be sufficient for most metabolites, while smaller effects can only be detected in larger samples of 1100–3000 subjects. With the aim of achieving such scale effects, large-scale epidemiological studies based on nutrimetabolomics findings have been widely conducted, varying from dietary assessment to metabolic profiling and risk prediction (see Table 3 for details).

### 3.3. Biofluid Samples

Since it is difficult to distinguish diet-induced changes in the metabolic profile from normal physiological changes, the type of biofluid sample should be carefully selected according to the required information and purpose. For the overall trend, the biofluid samples mainly used in the analysis were blood plasma or serum (*n* = 300; 56%) and urine (*n* = 169; 32%) (Figure 3b). In the early stages of development, studies using urine samples were predominant; however, since 2010, the number of studies using blood samples has increased significantly, and the proportion of studies reached 51% in 2010–2014 and 59% in 2015–2019; the proportions of studies using urine samples were 38% and 28% respectively, for these periods. In recent years, the number of studies on fecal samples (*n* = 41; 8%) has been growing, while human milk samples (*n* = 17; 3%) have remained steady and there have been only a few studies on saliva samples (*n* = 3; 1%).

#### 3.3.1. Blood (Plasma/Serum)

Recently, blood samples have become predominant compared to urine samples. Some of the obvious differences between blood and urine are the ratio of external to internal metabolites as well as concentration of non-nutrient substances [1,2]. Blood is rich in nutrients and metabolites being transported from one organ to another, and these metabolically active substances are sustained in the blood for a relatively long period of time, raising their blood levels. They are released into the urine only when the relevant renal threshold is exceeded. Due to this reason, the blood can contain a higher concentration of metabolically active compounds than urine. In addition, fat-soluble substances can be found in blood but not in urine. Therefore, as we can see in this survey, despite their invasiveness, blood samples are currently likely to be considered the most applicable biofluid to obtain robust metabolic signatures of dietary effects.

#### 3.3.2. Urine

The main aim of producing urine is to process unnecessary compounds in the body, and as a result, the concentration of internal metabolites as well as non-nutrients such as phytochemicals and compounds produced by chemical changes due to cooking [34] are likely to be higher in urine than in blood; thus, urinary samples are more suitable to assess these substances [1,2]. In addition, most of the urinary metabolites are excreted faster than the metabolites in the blood. In fact, the pioneering study by Walsh [16] compared diet-induced changes in metabolic profile among urine, plasma, and saliva, and only the urine showed a sensitive metabolic profile that reflected acute dietary intake. While urine can be obtained non-invasively, its handling may be complicated in terms of interpretation of metabolome variation because of the influence of spot urine and the difference in renal function. Several xenobiotics, medications, and medical conditions may also interfere with the metabolism and excretion of nutrients [35]. Due to these complexities, the number of studies remained below 30 per year throughout the entire period.

#### 3.3.3. Feces

Feces are promising biological samples to explore gut flora and its influence on dietary effects. In addition to its non-invasive sampling, the metabolic changes in blood and urine described above are likely to be associated with exogenous metabolites not only from digested and absorbed food-derived components but also from intestinal microbial flora metabolism, which is affected by diet and other health factors [4]. In fact, in our survey, studies using fecal samples were the third most common type of study after those using blood and urine samples, and as of 2019, the total number of researches by fecal samples reached almost two-thirds of those using urinary samples. That said, as the metabolites of the gut microbiota are not be directly related to the digestive and absorptive processes of the human body, the results should be analyzed in an integrated manner with those obtained by metabolomics of other biofluids such as blood and urine.

#### 3.3.4. Saliva

While saliva has not been widely used in the nutrimetabolomics field, it is expected to be a useful tool for monitoring changes in metabolic profile induced by diet [36]. This is because saliva is rich in oral microbial compositions as well as various hormones, which can provide important metabolic information [37,38]. This type of biofluid is also a non-invasive and easily collectable sample. Mounayar [39] reported that participants with high and low sensitivity to the taste of fat differed in salivary response to oleic acid. De Filippis [38] investigated whether eating habits could affect the formation of salivary bacteria flora and metabolomes among omnivore, ovo-lacto-vegetarians, and vegans, providing potential population-identifiable microbiota and metabolic markers.

#### 3.3.5. Human Milk

Nutrimetabolomics studies using human milk samples have been focused on because, in addition to being non-invasive and easy to collect, fetal and infancy nutritional statuses may affect their subsequent growth, health risk, or genetic modification. In fact, reports included not only studies on changes in perinatal milk composition, but also studies investigating links between maternal obesity and human milk metabolites [40] as well as risks of postnatal weight gain [41].

### 3.4. Fields of Application

The application fields of nutrimetabolomics are categorized into dietary assessment, metabolic profiling, risk prediction, gut microbiota diversity, genetic interaction, Human milk profiling, and diet sensitivities. The number and proportion of published studies were as follows: dietary assessment (*n* = 91; 20%), metabolic profiling (*n* = 191; 42%), risk prediction (*n* = 101; 22%), gut microbiota diversity (*n* = 30; 7%), genetic interactions (*n* = 7; 2%), human milk profiling (*n* = 15; 3%), and diet sensitivity (*n* = 17; 4%) (Figure 3c). Metabolic profiling, which explores biochemical changes caused by dietary intake, was predominant throughout the research period. On the other hand, there has recently been a significant growth in risk prediction, which investigates the direct impact on disease risks, and the number of these studies increased from 13% in 2010–2014 to 26% in 2015–2019 and ranked first among all application fields in 2019.

#### 3.4.1. Dietary Assessment

Studies on dietary assessment play a vital role in nutritional metabolomics, and a certain number of papers have been constantly reported. The focus of studies includes food compound intake biomarkers, food intake biomarkers, and dietary pattern biomarkers. Population-based nutritional studies, in which food intake is not accurately defined and controlled, have traditionally evaluated the nutritional status using practical self-reporting tools such as food intake frequency questionnaires (FFQs), dietary records, and 24-h recall. However, all of these have inherent limitations subjected to random and systematic errors [3]. Another issue in interpreting the findings of dietary intervention is compliance assessments. That is, how well participants adhered to the definition of the assigned diet, and if not, researchers cannot draw meaningful conclusions from the particular diet. However, these limitations caused by conscious or unconscious distortion can be overcome to some extent by objective assessment with metabolomics approaches. Altmaier [42] examined whether self-reported intake reflects de facto changes in metabolic profiling, showing the possibility of quality assessment of self-reports by comparing with metabolome data.

#### 3.4.2. Metabolic Profiling

Metabolic profiling with unknown biological consequences is one of the most promising ways to explore the potential health benefits of diet and has been the mainstream of nutrimetabolomics researches. The focus of studies includes effect biomarkers, which are indicators of response to a certain diet or dietary exposure of target function/biological response. Reports on this category have spanned a wide range of subcategories, especially focusing on food groups such as fruits, coffee/cocoa/tea, cereal/grains, dairy products, alcohol, and human/formula milk; dietary patterns such as energy-restricted diets, fasting, and wholegrain diet; and health risks such as metabolic syndrome (MetS), CVD, and cancer. These studies have been extensive and exploratory and are likely to correlate directly or indirectly with health risk and reach future disease risk prediction researches.

#### 3.4.3. Risk Prediction

Risk prediction in nutrimetabolomics aims to characterize the susceptibility to diseases induced by dietary factors. The focus of studies includes physiological or health status biomarkers, which reflect current risk of disease. The number of articles has been increasing notably from only three in 2010 to 38 in 2019. This likely means the field of nutrimetabolomics has entered the stage of clarifying the more direct role of dietary factors in the aspects of health and disease. The association between food and dietary patterns and disease risks has been mainly investigated in CVD, obesity, prediabetes/diabetes, and cancer. However, it is not easy to interpret the mechanism of identified risk markers and prove the causal relationship, as the effects of half-life of excretion and inter/intra-fluctuations are not yet fully clarified [10,43]. Thus, at this moment, the markers can be used to predict only specific aspects of individuals’ risks.

#### 3.4.4. Gut Microbiota Diversity

The gut microbiota interacts with dietary effects as a mediator between food intake and digestion and absorption into the human body, adding an even higher level of complexity to the overall picture of nutrimetabolomics. Therefore, in recent years, the focus of nutritional research is shifting from clarifying the direct effects of each dietary component to understanding the comprehensive dietary effects such as interactions with the microbiota or host microbial ecology, which indirectly affects metabolic changes. The focus of studies includes effect biomarkers and susceptibility biomarkers. Indeed, the structure and composition of the fecal microbiome have been shown to be closely associated with both human health and diseases [44]. Recent nutrimetabolomics efforts include the contribution of gut microbiota to host metabolism in vegetarians [45], differential effects of typical Korean versus American-style diets on gut microbial composition and metabolic profile [46], and the impact of dietary fiber supplementation on modulating microbiota-host-metabolic axes in obesity [47].

#### 3.4.5. Genetic Interaction

Studies of this field focus on susceptibility biomarkers such as host factor biomarkers. Despite the small number of research reported, understanding the interrelationships of genotypes with metabolic changes is indispensable for precision nutrition in identifying individuals who may benefit the most from specific dietary strategies. This is because genetic interaction can affect a variety of biochemical processes such as nutrient digestion and absorption, metabolism, turnover, and excretion [48]. For example, Lai [49] elucidated the interaction of the apolipoprotein A-II (APOA2) genotype with saturated fat uptake, which is a risk factor for obesity, demonstrating the link between epigenetic status and metabolic networks. Kakkoura [48] also found the interactions of nine breast cancer-related polymorphisms with the dietary pattern of the MED. In a study of high glucoraphanin broccoli on cancer prevention [50], the plasma metabolite profiles were adjusted for the interaction of the Poly(A) polymerase gamma (PAPOLG) genotype affecting mitochondrial function, fixing inconsistencies of the results of study.

#### 3.4.6. Human Milk Profiling

As mentioned above, human milk is not only an essential source of nutrition for infants but also an indispensable biofluid for providing information on maternal health as well as predicting health risks for infants’ subsequent growth during lactation and after weaning. The focus of studies includes effect biomarkers and host factor biomarkers. The research exploring human milk composition is not a new field, but comprehensive metabolomics approaches provide deeper insights into the nutritional requirements of developing infants, including the influence of gestational age, disease and its treatment, and the mother’s habitual diet.

#### 3.4.7. Diet Sensitivity

Since dietary biochemical reactions are multifaceted in our body, the targeted application fields have widened from direct effects of food consumption itself to sensitivity and preferences in diet as well as its sensory effects. The focus of studies includes individual variability biomarkers and biomarkers of phenotypic traits. These distinctive metabolomics studies included influences of anxiety traits and microbiota in dietary preferences [30,51], taste perception phenotype in sensitivity to taste of fat [39], food preference patterns in a twins’ cohort [52], and cognitive and hedonic responses to meal ingestion [53]. Details of these topics are shown in Table A3.

### 3.5. Dietary Factors

Dietary factors are classified into nutrients, food groups, and dietary patterns (Figure 3d). Throughout the whole period, the most commonly reported were food groups (*n* = 204; 48%), followed by dietary patterns (*n* = 125; 29%) and nutrients (*n* = 99; 23%). Studies on dietary patterns showed a significant upward trend from 20% in 2010–2014 to 33% in 2015–2019, while food groups decreased from 56% to 44%, yet remained top-ranked. The detailed trends of the dietary factors are shown in Figure 4.

#### 3.5.1. Nutrients

While the number of articles in this category has not increased so much recently, studies of how individual nutrients affect metabolic changes are basic nutritional approaches to clarifying the causal relationship between health and diet. Lipids (*n* = 25; 25%), vitamins (*n* = 20; 20%), and non-nutrients (*n* = 24; 23%) were the main research categories (Figure 4a). In particular, there has been a marked increase in the number of studies on non-nutrition over the past five years.

For lipids, reports of *ω*-3 fatty acids have been the mainstream, constituting 16 of 25 total articles in this category. The impact of trans fatty acids (TFA) related to CVD and the mechanisms of TFA-induced disease were also explored, revealing elevated levels of specific polyunsaturated long-chain phosphatidylcholines and a sphingomyelin [54].

Regarding vitamins, vitamin D, which can have multifaceted effects in the prevention of risks such as osteoporosis, MetS, and atherosclerosis, has constantly been the main topic in the category. O’Sullivan [55] suggested that responsiveness to vitamin D supplementation in terms of MetS was mediated in part through modulation of lipid metabolism. Fernández-Arroyo [56] also investigated the effect of vitamin D on the postprandial lipidomic profile in obese patients to clarify the mechanism underlying the improvement of CVD risk.

The recent increasing category was non-nutrients, including bioactive compounds, mainly focusing on the dietary effects and its metabolic mechanisms of phytochemicals such as polyphenols and flavonoids. Edmands [57] investigated 6 polyphenol-rich foods (coffee, tea, red wine, citrus fruit, apples, pears, and chocolate), identifying more than 80 polyphenol metabolites associated with the selected foods. In addition, various interventions using polyphenols have also been reported, including rice bran [58], a seaweed polyphenol extract [59], bean genistein [60], pomegranate urolithin [61], limonene in citrus peel [62], and secoiridoids from a seed/fruit extract [63]. Another non-nutrient of interest was trimethylamine-*N*-oxide (TMAO), which can be an indicator of increased CVD risk from meat and dairy intake [64,65], although the levels may also be individually associated with fish intake [66].

#### 3.5.2. Food Groups

A wide range of food groups have been reported from the early stage of the nutrimetabolomics field. Among them, studies related to fruit (*n* = 31; 15%), coffee/cocoa/tea (*n* = 28; 14%), alcohol (*n* = 21; 10%), human/formula milk (*n* = 18; 9%), and cereal/grains (*n* = 16; 8%) have been continuously reported over the period (Figure 4b).

The associations of metabolite levels with extensive food groups have been investigated [67]. Guertin [5] identified a total of 39 serum biomarkers of 36 food groups, such as peanuts with tryptophan betaine, coffee with trigonelline-*N*-methylnicotinate and quinate, and alcohol with ethyl glucuronide. Playdon [68] also reported the difference of metabolomic markers between serum and urine, associated with 46 food groups.

Studies reporting other remarkable food groups such as fruits and coffee/cocoa/tea, which are rich in phytochemicals, have also provided significant clues to the association between food intake and health risks. For example, proline betaine in citrus fruits has beenregarded as one of the most reliable metabolic signatures among potential food mark-ers [69,70]. Many studies focusing on berries and the effects of reducing metabolic risk [71,72] have also been reported. As for coffee, trigonelline, caffeine and its deriv-atives, and quinic acid can be highly correlated with its intake [73,74].

Regarding cereal and grains, studies of wholegrains have been continuously reported since 2010. While epidemiological findings have constantly supported the fact that wholegrains reduce the risks of chronic disease and cancer, the basic mechanism of the health effects has not been elucidated. Therefore, metabolic changes by wholegrain wheat, which characterizes the MED [75], and wholegrain rye, which characterizes the healthy Nordic diet [76], have been chiefly investigated.

Since meat intake may increase the risk of chronic diseases such as MetS and various types of cancer, nutrimetabolomic approaches would be crucial to understand its biological effects. Cuparencu [77] identified nine red meat, four white meat, and eight common meat biomarker candidates, from collagen degradation, flavor compounds, and amino acid metabolism. Cheung [66] also found potential markers including anserine pecified for chicken intake, as well as carnosine and three acylcarnitines for meat intake in general. As meat processing methods may also have cancer risks, Wedekind [34] investigated the effects of smoked meat intake (e.g., bacon, salami, and hot dogs) and identified four syringol sulfates as its biomarkers. A study by Wei [78] highlighted metabolic signs of host-microbiota interaction in meat intake supporting the role of gut microbiota.

#### 3.5.3. Dietary Patterns

Due to the recent expectation that dietary patterns can play a major role in the effects on health [32,79], there has been a particularly increasing number of research reports applying metabolomics to elucidate the association between dietary patterns and metabolic changes. In 2019, the number of articles (*n* = 41) was almost double that of the previous year (*n* = 22), and it was ranked first in the application fields. While the number of MED-related research, which attracted early attention as a healthy diet [32], was the highest overall (*n* = 15; 12%), various derivative patterns have recently been advocated, and the range of research has expanded (Figure 4c). Typical other dietary patterns include plant-based diet (vegetarian/vegan diet; *n* = 10; 8%), wholegrain/low-glycemic index diet (*n* = 9; 7%), the New Nordic diet (*n* = 6; 5%), the Dietary Approaches to Stop Hypertension (DASH) diet (*n* = 4; 3%), the Health Eating Index (HEI)/alternate Health Eating Index (aHEI) (*n* = 3; 2%), and the WHO Healthy Diet Indicator (HDI) (*n* = 2; 2%).

Healthy dietary patterns that comply with national dietary guidelines are especially expected to have positive health effects with clear evidence. Playdon [79] examined the correlation between serum metabolomics and dietary patterns based on HEI-2010, alternate MED (aMED), HDI, and Baltic Sea Diet (BSD), showing that lysolipids and plant xenobiotic pathways are most strongly associated with dietary quality. McCullough [80] also identified serum metabolomic markers including metabolites such as sphingomyelin, hydroxy-CMPF, *β*-cryptoxanthin, and docosahexaenoate, to discriminate four dietary patterns based on the aMED, aHEI, HEI, and DASH.

While it may be conceivable to recommend a wholegrain and low glycemic load diet [81,82] with more vegetables, the application of an entirely plant-based diet is still being discussed. To provide a proper basis for this argument, recent studies [83,84,85] analyzed vegan, vegetarian, and omnivore subjects and showed that different metabolic patterns could discriminate between animal foods diets and plant-based diets.

Extensive comparisons of metabolite profiling have revealed that metabolite changes are affected by region-specific dietary habits. Studies involved a comparison of urine metabolites of northern and southern Chinese populations [86], plasma and urine metabolites of European populations [87], and fecal metabolites of Korean and American-style diet populations [46]. Nutrimetabolomics was also applied to predict health risks in poverty-stricken areas. An investigation of urine metabolic changes among people at risk-of-poverty in European populations provided biomarkers of undernutrition [88].

### 3.6. Targeted Health Risks

Recently, nutrimetabolomics studies have focused more on specific metabolites and pathways to prevent the development of health risks and their complications. The trend can lead to elaborate subclassification of health risks and personalized nutrition according to individuals’ metabolic characteristics. Thus, the range of targeted risks has been expanding annually. Throughout the survey period, lifestyle-related diseases risks were mainly reported, including prediabetes/diabetes (*n* = 37; 15%), obesity (*n* = 24; 10%), MetS in general (*n* = 18; 8%), and hypertension (*n* = 10; 4%). The numbers of studies targeting CVD (*n* = 35; 15%), cancer (*n* = 19; 8%), and maternal and pediatric health (*n* = 26; 11%) have also increased (Figure 4d).

As diabetes and CVD are chronic diseases that can be closely associated with dietary factors, many nutrimetabolomics studies have been reported. Meyer [89] provided a model to predict insulin sensitivity improvements using baseline blood metabolomics, after a 6-month-low-calorie diet in overweight subjects. Zheng [90] suggested that a sugar-rich dietary pattern may be partially attributed to oxidative stress and disordered lipidomic profiles when analyzing metabolic profiles of African Americans with CVD risks.

Specific food intake and dietary patterns are regarded as potential preventive or risk factors of various cancers [91]. For instance, consuming foods, such as whole-grain rye [92,93], coffee [74], navy beans [93], and black raspberries [94], induced metabolomic changes, preventing prostate, colorectal, and lung cancers. Playdon [95] investigated the relationship between 55 food groups and breast cancer risk, revealing that 3 specific metabolites were associated with overall breast cancer risk, and 19 were associated with estrogen receptors. A study on prostate cancer risk by Beynon [96] also reported that dietary lycopene decreased serum levels of pyruvate, whose elevation can be causally related to the cancer risk. Dietary patterns that affect the cancer risks and its metabolic changes were also investigated by several studies, including the relationship between genetic interactions with MED and breast cancer risk [48] and between energy-restriction diet and metabolic changes reducing gene expression associated with breast cancer [31].

### 3.7. Future Aspects and Issues

Nutrimetabolomics can contribute to the application of precision nutrition for the management of diets and risks in the future, as the field has the possibility to predict individual metabolic dynamics with objective assessments. As surveys of individual nutrients and food groups are not sufficient to unravel the causal relationship between complex eating behaviors and health risks, focus should be placed on research of dietary patterns describing comprehensive dietary habits. Further, in recent years, the field of nutrimetabolomics has entered the stage of clarifying the direct relationship between diet and disease risks. Therefore, it is crucial to elucidate interactions with genotypes and its epigenetic effects, and with the gut microbiota. Further investigation of diet sensitivity and sensory effects on metabolic changes are also expected.

For this new field to be positioned in the mainstream of nutritional science as practical indicators, several issues need to be overcome including validating and reproducing candidate metabolites, causal verification, elucidation of inter/intra-fluctuations, half-life duration of excretion [97], methodologies in study design, and advanced data analysis. For instance, although non-targeted approaches that enable exploratory research are more suitable to detect unknown compounds and generate new hypotheses, the high cost of advanced devices, densification of acquired data, and complex statistical analysis are required. On the other hand, targeted approaches follow conventional hypothesis-driven research, but are suitable for quantification. Landberg [97] suggested that ideally, for the detected set of metabolic signatures to be practically applicable to dietary biomarkers, targeted and non-targeted approaches should complement each other.

Systematic approaches to define and classify biomarkers due to standardized criteria and their intended use were also critical for validation of the markers as applicable assessment tools [98,99,100]. The Hohenheim Conference’s consensus statement on the definition of biomarkers in nutrition defined “test results related to exposure, susceptibility or biological effects” [101]. The joint programming initiative “Healthy Diet for a Healthy Life” (JPI-HDHT) under Food Biomarker Alliance (FoodBAll), which aims to support the systematic evaluation of new biomarkers, proposed a classification scheme for biomarkers [101] dividing nutritional biomarkers into food intake or nutritional status (recent or long-term) biochemical indicators, nutritional metabolism indicators, and biochemicals for food intake. To describe in detail the dietary and health field, Gao [12] suggested an improved scheme for dietary biomarker classification consisting of exposure biomarkers, effect biomarkers, and susceptibility biomarkers, with six subclasses.

### 3.8. Limitations

As the purpose of this scoping review was to provide an overview of development trends from a broad perspective over a long period of time, we used the single data source of MEDLINE, which is the most practical database in the biomedical field in terms of comprehensiveness and extraction efficiency. However, based on this scoping review, it is essential to make thorough searches from multiple data sources when conducting systematic reviews with qualitative evaluation and comprehensiveness in further researches.

In addition, since this scoping review mainly aimed to exploratorily examine the extent, variety, and characteristics of the evidence on the application fields of nutrimetabolomics with a visual representation of results, we did not critically appraise a cumulative body of evidence. Thus, it is indispensable to consider these evaluations in order to strengthen the levels of the evidence for future in-depth reviews focusing on individual application areas.

Since the years 2020 to 2021 were affected by COVID-19, the past 20 years up to 2019 were surveyed in order to ensure the continuity for tracking the annual transition of the number of publications. Furthermore, although the relevant papers published in journals are mostly covered, the so-called “gray papers” were not included, and the publication bias may also have curbed the number of studies that had conclusions that were not expected. However, we believe that our initial goal of overviewing developments of the nutrimetbolomics from a long-term perspective has been achieved.

## 4. Conclusions

Nutrimetabolomics, which can elucidate unknown mechanisms in the relationship between diet and health, has been rapidly gaining attention as one of the essential omics approaches. This scoping review outlined the transition of development, recent focuses, and future aspects of this new field from the early stages of the development to the present over the last two decades. The growth trend of the field was proven by the remarkable expansion in the number of articles. Among them, objective dietary assessment, metabolic profiling, and disease risk prediction were positioned as three of the principal applications of nutrimetabolomics. Particularly, there have been signs of surges in studies applying the dietary pattern and the disease risk prediction. Investigations of the relationship with gut microbiota diversity, genetic interactions, and diet sensitivity will also be expected research areas. The accumulation of these advances will enable nutrimetabolomics to contribute to personalized nutrition in the future.

## Figures and Tables

**Figure 1 nutrients-13-03760-f001:**
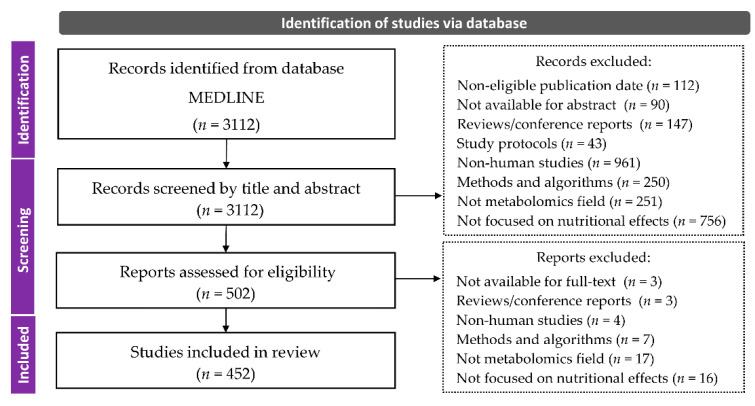
Flow diagram of search and article selection.

**Figure 2 nutrients-13-03760-f002:**
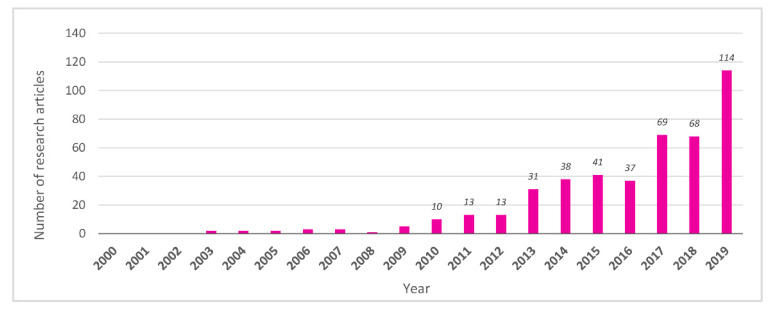
Total number of research articles in nutrimetabolomics (focused on human study).

**Figure 3 nutrients-13-03760-f003:**
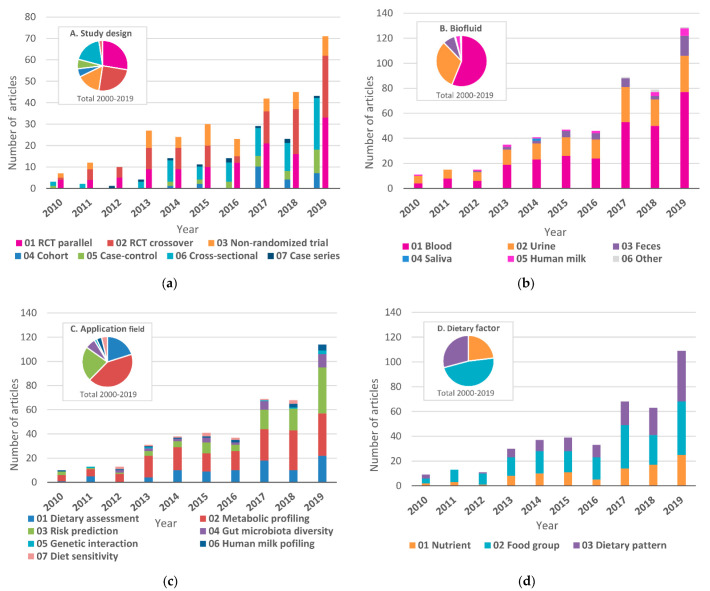
Number of published articles for the main categories of nutrimetabolomics: (**a**) study design (*n* = 456); (**b**) biofluid (*n* = 534); (**c**) application field (*n* = 452); (**d**) dietary factor (*n* = 428); Studies are categorized by the main subject described in the article, and are placed in multiple categories when multiple items are the main target. The aggregated results are shown in Appendix A.

**Figure 4 nutrients-13-03760-f004:**
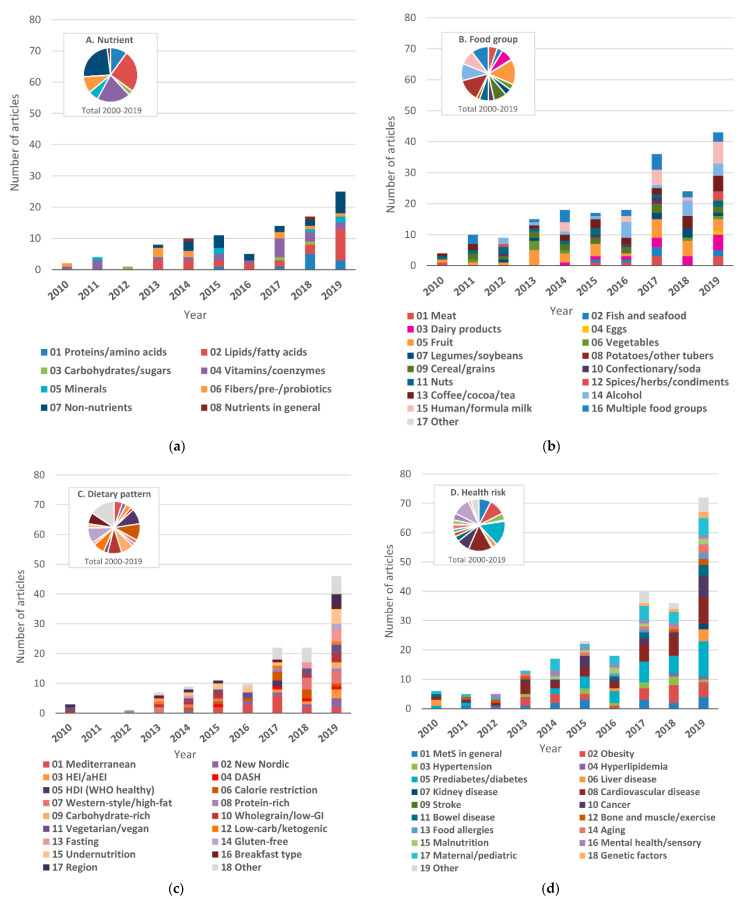
Number of published articles for the subcategories of nutrimetabolomics: (**a**) nutrient (*n* = 100); (**b**) food group (*n* = 206); (**c**) dietary pattern (*n* = 130); (**d**) targeted health risk (*n* = 239); Studies are categorized by the main subject described in the article, and are placed in multiple categories when multiple items are the main target. The aggregated results are shown in Appendix A. HEI, Health Eating Index. aHEI, alternate Health Eating Index. HDI, Healthy Diet Indicator. DASH, Dietary Approaches to Stop Hypertension.

**Table 1 nutrients-13-03760-t001:** Literature search strategy ^1^.

Medline Search	
Search engine	PubMed
Keywords ^1^ Search formula	(metabolomics OR metabonomics) AND (nutrition OR food OR diet OR meal OR intake OR consumption)
Species	Humans
Publication date	2000–2019
Publication type	Excluding: review/systematic review

^1^ Keywords were converted to PubMed-defined transition terms in the database search formula.

**Table 2 nutrients-13-03760-t002:** Pioneering human studies of the nutrimetabolomics (2000–2009).

Year	Author	Research Focus	Design ^1^	*n* ^2^	Sex	Biofluid ^3^	Method ^4^	Ref.
2003	Lenz, et al.	Biofluid comparison	NRCT	12	M	U, P	NMR	[14]
	Solanky, et al.	Isoflavone intake	NRCT	5	F	P	NMR	[22]
2004	Teague, et al.	Alcohol (ethyl glucoside) consumption	NRCT	2	FM	U	NMR	[24]
	Lenz, et al.	Diurnal fluctuation/regional difference	CSR/CS	30/120	FM	U	NMR	[15]
2005	Wang, et al.	Chamomile tea consumption	NRCT	14	FM	U	NMR	[17]
	Solanky, et al.	Isoflavones intake	NRCT	9	F	U	NMR	[25]
2006	Van Dorsten, et al.	Green tea/black tea consumption	RCT-CO	17	M	U	NMR	[18]
	Stella, et al.	Meat diet/vegetarian	RCT-CO	12	M	U	NMR	[26]
	Walsh, et al.	Biofluid comparison	NRCT	30	FM	U, P, SV	NMR, MS	[16]
2007	Rezzi, et al.	Dietary preferences	RCT-CO	22	FM	U, P	NMR	[27]
	Bertram, et al.	Milk/meat protein for child nutrition	RCT-P	24	M	U, S	NMR	[28]
	Walsh, et al.	Phytochemical intake	NRCT	21	FM	U	NMR, MS	[23]
2008	Law, et al.	Data comparison between different analytical methods	NRCT	8	M	U	NMR, LC-MS, GC-MS	[29]
2009	Martin, et al.	Dietary preferences and anxiety trait	RCT-P	30		U, P	NMR, MS	[30]
	Stalmach, et al.	Coffee consumption	NRCT	11	FM	U, P	LC-MS	[19]
	Llorach, et al.	Cocoa consumption	RCT-CO	10	FM	U	LC-MS	[21]
	Ong, et al.	Energy restriction on breast cancer	RCT-P	19	F	U, S	GC-MS	[31]
	Altmaier, et al.	Coffee consumption	CS	284	M	S	LC-MS, MS	[20]

^1^ RCT-P: randomized controlled trial-parallel, RCT-CO: RTC-crossover, NRCT: non-randomized clinical trial, CS: cross-sectional, CSR: case series. ^2^ Number of subjects in the study. ^3^ P: plasma, S: serum, U: urine, SV: saliva. ^4^ NMR: nuclear magnetic resonance, MS: mass spectrometry, LC: liquid chromatography, GC: gas chromatography.

**Table 3 nutrients-13-03760-t003:** Large-scale epidemiological studies with nutrimetabolomics focus (2000–2019).

Large-Scale Epidemiological Study	Population	Nutrimetabolomics Focus
Alpha-Tocopherol, Beta-Carotene Cancer Prevention study (ATBC)	Finland	Beta -carotene (2013), vitamin D (2016), diet indexes (2017)
Atherosclerosis Risk in Communities Study (ARIC)	USA	Dietary habits among African Americans (2014), alcohol (2016)
Cancer Prevention Study-II Nutrition Cohort (CPS- II Nutrition)	USA	Food group (2018), dietary indexes (2019)
Cardiovascular disease, Living, and Ageing in Halle (CARLA)	Germany	Effects of fasting time (2018)
Cooperative Health Research in the Region Augsburg (KORA)	Germany	Self-reported dietary habits (2011), fecal sterols (2019)
European Prospective Investigation into Cancer and Nutrition (EPIC)	10 European countries	Dietary pattern (2013, 2015, 2017), wholegrains (2014), meat/fish (2015,2017), alcohol (2018, 2019), coffee (2019), smoked meat (2019)
Finnish Dietary, Lifestyle, and Genetic Determinants of Obesity and Metabolic Syndrome (DILGOM)	Finland	Food neophobia (2019)
International Study on Major Nutrients and Micronutrients and Blood Pressure (INTERMAP)	UK, USA, China, Japan	Phenotype diversity (2008), fruit/proline betaine (2010), Chinese population (2010), African Americans (2013), WHO healthy (2019)
Nurses’ Health Study (NHS)	USA	Branched-chain amino acids (2018), nuts (2019)
Prevención con Dieta Mediterránea (PREDIMED)	Spain	MED effects (2015), CVD risk (2016, 2017), nuts (2014), pulse (2017), coffee/cocoa (2015, 2019), red wine (2019), choline pathway (2017)
Special Turku Coronary Risk Factor Intervention Project (STRIP)	Finland	Dietary counseling (2018)
STORK-Groruddalen cohort study (STORK)	Norway	Breastfeeding (2014)
Systems biology in Controlled Dietary Interventions and Cohort Studies (SYSDIET)	5 Nordic countries	Healthy Nordic diet (2019)
TwinsUK Study (TwinsUK)	UK	Food preference (2015), self-reporting (2016), dairy (2017), omega-3 fatty acid (2017), gut microbiota (2017)

MED, Mediterranean diet. CVD, cardiovascular disease.

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
