# Peer review of "A Scoping Review of the Application of Metabolomics in Nutrition Research: The Literature Survey 2000–2019"

_nutrients, 2021, doi:10.3390/nu13113760_

Round 1

Reviewer 1 Report

Although the subject of this review is interesting, there are methodological errors that detract from the results provided.

Among others:

  • According to PRISMA, the complete final search equation must be included and is not indicated in this review.
  • In addition, it does not indicate whether the search terms have been searched in the MeSH registration fields or also in the title and abstract fields.
  • Only one bibliographic database, MEDLINE, has been interrogated, which can lead to a very important lack of information. It would have been desirable to interrogate others such as Embase, Web of Science, etc.
  • No type of evaluation of the documentary quality has been carried out and the possibility of the existence of biases has not been studied either.
  • The level of evidence and recommendation provided by this review is not indicated.

Reviewer 2 Report

The subject of the review was an analysis of research papers on nutrient metabolomics that focus on the interaction between diet with health, from 2009 to 2019. The authors used the MEDLINE database as a source of articles. Initially, they selected over three thousand works, then their number was narrowed down to 450 that met the selection criteria.  These criteria have been clearly specified, the reasons for exclusion have been adequately explained. I have no objections to the method of extraction and the criteria used.

The analysis focused on the following areas: application (e.g. metabolic profiling), dietary factors (such as nutrient group or specific health risk), design of study, type of samples. In my opinion, the review is very well thought out. It takes into account the most important problems of research on metabolomics of nutrients. The description of the analyzed works shows that the authors used the correct selection algorithm. They present valuable research that makes a significant contribution to understanding the relationship between nutrition and health outcomes.

Nutrimetabolomic research has been carried out over the last 10 years. The authors indicated the most important applications and the latest trends (including the genetic profile).

All the text is in correct English, the structure is simple and clear. I did not find any linguistic errors. Figures and tables are indicative and clear.

Reviewer 3 Report

See attached document for details.

Round 2

Reviewer 1 Report

It is true that the review has improved. But, there is still a methodological problem:

This work is a bibliographic review but it cannot be considered a systematic review sensu stricto. Therefore, it could be accepted, as long as the authors delete the paragraph from lines 76 to 78 "The 76 PRISMA-ScR (Preferred Reporting Items for Systematic Review and Meta-Analysis, ex- 77 tension for Sscoping Rreviews extension) guideline was followed [12, 13]." As well as any reference to the fact that it is a systematic review (for example, in line 553)
